# Repurposing Metabolic Inhibitors in the Treatment of Colon Adenocarcinoma Patient-Derived Models

**DOI:** 10.3390/cells12242859

**Published:** 2023-12-18

**Authors:** Bora Lee, ChuHee Lee, Hae-Min Moon, Se-Young Jo, Se Jin Jang, Young-Ah Suh

**Affiliations:** 1Department of Biomedical Sciences, Asan Medical Center, The University of Ulsan College of Medicine, Seoul 05505, Republic of Korea; leebora74@hotmail.com (B.L.); glorymhm_@naver.com (H.-M.M.); shockyoung@yuhs.ac (S.-Y.J.); 2Department of Biochemistry and Molecular Biology, School of Medicine, Yeungnam University, Daegu 38541, Republic of Korea; chlee2@ynu.ac.kr

**Keywords:** cancer metabolism, biguanide, colon tumor, tumor organoid, patient-derived xenograft

## Abstract

The effect of agonists on AMP-activated protein kinase (AMPK), mainly metformin and phenformin, has been appreciated in the treatment of multiple types of tumors. Specifically, the antitumor activity of phenformin has been demonstrated in melanomas containing the v-Raf murine sarcoma viral oncogene homolog B1 (BRAF) activating mutation. In this report, we elucidated the synergistic antitumor effects of biguanides with metabolism inhibitors on colon tumors. Phenformin with 2-deoxy-D-glucose (2DG) inhibited tumor cell growth in cancer cell lines, including HT29 cells harboring BRAF- and p53-mutations. Biochemical analyses showed that two chemotherapeutics exerted cooperative effects to reduce tumor growth through cell cycle arrest, apoptosis, and autophagy. The drugs demonstrated activity against phosphorylated ERK and the gain-of-function p53 mutant protein. To demonstrate tumor regressive effects in vivo, we established patient-derived models, including xenograft (PDX) and organoids (PDO). Co-treatment of biguanides with chemotherapeutics efficiently reduced the growth of patient-derived colon models in comparison to treatment with a single agent. These results strongly suggest that significant therapeutic advantages would be achieved by combining AMPK activators such as phenformin and cancer metabolic inhibitors such as 2DG.

## 1. Introduction

Colorectal cancer (CRC) represents the third leading cause of cancer-related death worldwide [1]. The treatment for CRC patients is mainly surgery, followed by adjuvant treatments of chemotherapy and radiotherapy. Although targeted therapy and techniques for molecular analysis and early detection have developed incredibly in the last decade, strategies for CRC treatment are still in demand because of the complex response of individuals to treatment.

Not only enhanced tumor metabolism but also its specific microenvironment are recognized as a unique aspect in multiple tumor types, and the antitumor effects of biguanides, metformin and phenformin, accumulate when re-evaluated in several tumors [2,3]. The antitumor effects of phenformin have been recently elucidated in multiple cancer types [4,5,6,7,8,9], including chemo-resistant rectal cancer [10]. Numerous reports regarding the antitumor effects of phenformin have emphasized its potent ability to reduce cancer cell proliferation, with a smaller dose than used for metformin; therefore, phenformin has been highlighted as a candidate for cancer therapeutics [2]. Evidence demonstrating that metformin and phenformin inhibit cell proliferation, angiogenesis, and tumor growth has been collected for various cancers, including breast cancer, prostate cancer, lung cancer, melanoma, and ovarian cancer. These drugs are known to activate energy-sensing AMP-dependent protein kinase (AMPK) and to inhibit complex-I in the mitochondrial electron transport chain [4,7,11]. Moreover, clinical studies have demonstrated reduced tumor development and increased survival rates for patients suffering various types of cancer, including breast, pancreatic, lung, and prostate cancers, when metformin has been administered [12,13]. Metformin and phenformin inhibited epithelial–mesenchymal transition (EMT) and induced the apoptosis of chemoresistant CRC cells [10]. The synergistic effects of biguanides in combination with chemotherapeutics or radiation treatment have also been reported [14,15,16,17,18].

The anticancer effect of glycolysis inhibitor 2-deoxy-D-glucose (2DG) has been emphasized for various types of cancers, both as a single agent or in combination with chemotherapeutics [19,20]. The C-2-hydroxyl group of glucose was modified using hydrogen, competing with glucose for uptake into cells and resulting in cytotoxicity, starving cancer cells of energy. Moreover, the effect of 2DG on colon cancer cells when used in combination with phenformin was found to not only enhance the inhibition of cell growth but also reverse the acidification caused by phenformin treatment [21], providing evidence supporting the development of phenformin as a promising cancer therapeutic.

To evaluate antitumor efficacy precisely, one of the most important factors is model fidelity. Patient-derived models such as xenograft and organoids are therefore incredibly valuable not only for preclinical but also for translational research. As such, the development of bio-banking systems comprising tumor organoids originating from colon patients has filled the gap between cancer genetics and patient trials and has enabled the analysis of genotype-to-phenotype correlations [22].

It is also known that the activation of AMPK can be hindered by the gain-of-function mutant p53 protein. Previously, we observed that phenformin and 2-deoxy-D-glucose reduced tumor growth in syngeneic mice harboring the p53 gain-of-function mutation, indicating that destabilizing the oncogenic mutant p53 protein is a promising strategy in order to kill tumor cells exhibiting oncogene addiction [23]. In the present study, we evaluated the effects of phenformin and 2-deoxy glucose on colon tumors in vitro and in vivo using cancer cell line, xenograft, and patient-derived models such as PDX and tumor organoids.

## 2. Materials and Methods

### 2.1. Cell Culture and Reagents

We obtained human colorectal cancer cell lines HT29, SW620, and RKO from the American Type Culture Collection (ATCC) (Manassas, VA, USA). Cells were cultured using Dulbecco’s modified Eagle’s medium or RPMI-1640 medium (GIBCO BRL, Carlsbad, CA, USA) containing 10% fetal bovine serum (GIBCO BRL) and antibiotics (100 U penicillin, 100 μg/mL streptomycin) in an incubator at 37 °C with 5% CO_2_ supplement. All cell lines were tested routinely for negative mycoplasma using PlasmoTest (InvivoGen, San Diego, CA, USA) to avoid common cell culture contamination, and cell viability was verified by performing a trypan blue exclusion assay. Whenever needed, cells were counted on an Automated Cell Counter (BioRad, Hercules, CA, USA). Phenformin and 2-deoxyglucose (2-DG) were purchased from Aldrich-Sigma Chemical Co. (St. Louis, MO, USA). Antibodies to cyclin D1, Caspase-3, poly (ADP-ribose) polymerase (PARP) and phospho-ERK1/2 were obtained from Cell Signaling Technology (Danvers, MA, USA), and antibodies to β-actin were purchased from Sigma-Aldrich (St. Louis, MO, USA). Antibodies for p53 and ERK1/2 were obtained from Santa Cruz Biotechnology (Santa Cruz, CA, USA). Horseradish-peroxidase-conjugated goat anti-mouse or goat anti-rabbit secondary antibodies were purchased from Cell Signaling Technology.

### 2.2. Organoid Culture

The establishment of patient-derived colon cancer organoids was performed in accordance with the Ethics Committee of the Asan Medical Center (Seoul, Republic of Korea) and under the institution’s guidelines, with patients’ informed consent, as described in previous reports [22,24]. Briefly, surgically resected tumor tissues (1–4 cm^3^) from colorectal cancer patients were washed and cut into small pieces in cold Hank’s balanced salt solution (HBSS) with 1% penicillin/streptomycin (Gibco, Basel, Switzerland). Tumor tissues were then incubated with collagenase II (1.5 mg/mL, Roche, IN, USA) at 37 °C for 2 h with occasional shaking. Large fragments from the incubated suspension were separated via passing through a 100 μm cell strainer (BD Falcon, CA, USA), and cells were harvested through centrifugation at 1000 rpm for 3 min. The cell pellets were re-suspended in colorectal tumor organoid culture medium (DMEM/F12 medium (Invitrogen, Carlsbad, CA, USA), 50 ng/mL human EGF (Invitrogen), N2 (Invitrogen), B27 (Invitrogen), 1 mM N-acetylcysteine (Peprotech, NJ, USA), 10 mM nicotinamide (Peprotech), 10 nM gastrin I (Peprotech), 500 nM A83–01 (Peprotech), 10 μM ROCK inhibitor (Peprotech), and 1% penicillin/streptomycin (Gibco)) and were solidified with Matrigel (Corning, Corning, NY, USA) on a culture plate. The medium was changed every 4 days, and the organoids were passaged after 2–3 weeks. The organoids were harvested, washed with cold DPBS, and incubated with TrypLE Express (Invitrogen) for 10 min at 37 °C for further cultivation. Dissociated pellets were again washed and reseeded at 1:3 ratios onto the new plates. For cryopreservation, organoids harvested from Matrigel were suspended in freezing media (Culture Media 7: ES grade FBS (Gibco) 2: DMSO (Sigma) 1) and were stored in a nitrogen gas tank.

The morphology of colorectal organoids was evaluated through H&E staining. Cultured organoids were fixed in 4% paraformaldehyde (PFA) followed by dehydration, paraffin embedding, sectioning, and standard H&E staining.

### 2.3. Protein Expression Analysis

Western blotting was carried out to analyze the expressed proteins as described elsewhere. Total cell proteins (15~20 μg) were separated on SDS-PAGE gels and transferred to a membrane to be analyzed. Primary antibodies were applied to membranes overnight at 4 °C and were detected with the corresponding horseradish-peroxidase-conjugated goat anti-mouse or goat anti-rabbit secondary antibody. Proteins were then visualized using an Enhanced ChemiLuminescence kit or LAS4000 (GE Healthcare Bio-Sciences, Chicago, IL, USA).

### 2.4. Generation of Patient-Derived (PDX) Models

All procedures to establish patient-derived models, such as PDX and tumor organoids, were approved by the Asan Medical Center Institutional Review Board (IRB# 2012-0112) and Asan Medical Center Institutional Animal Care and Use Committee. Experimental protocols were designed according to the Asan Medical Center institutional guidelines for the care and use of animals. All patient tumor tissues were obtained with informed consent in accordance with the ethical standards of the Asan Medical Center Institutional Research Committee. The establishment of PDX and tumor organoids are as described in previous reports [25,26,27]. Briefly, a piece of tumor tissue in 3 × 3 × 3 mm^3^ size was implanted subcutaneously into the flank of 5-week-old NOD-SCID mice (Charles River Laboratories, Wilmington, MA, USA) to generate PDX. After 2–3 months, when the tumor size reached more than 1 cm^3^, the tumors were harvested and transplanted into male athymic nude mice (BALB/c-nude; 5 weeks old; Central Lab. Animal Inc., Seoul, Republic of Korea) [25,26,27] for further propagation and experiments.

### 2.5. Assessment of Tumor Growth Inhibition In Vivo

All procedures for colorectal xenograft/PDX experiments were approved by the Asan Medical Center Institutional Animal Care and Use Committee and performed as described in previous reports [25,26,27]. Briefly, for xenograft experiments, HT29 cells (2 × 10^6^) suspended in 200 μL of PBS were inoculated subcutaneously into the flank of five-week-old male athymic nude mice purchased from Central Lab. Animal Inc. When the tumors grew to approximately 100 mm^3^ in size (14–21 days), the mice were treated with vehicle or 2-DG plus phenformin (750 mg/kg of 2-DG and 100 mg/kg of phenformin) orally once per day for the duration of the experiment, randomly divided into three groups. The mice numbered five for untreated groups and eight for treated groups with 2-DG plus phenformin.

Tumor growths were monitored for 25 days after the onset of treatment by measuring with digital calipers daily, followed by conversion to tumor volumes using the following formula: W1 × W2 × W2/2 = x mm^3^ (where W1 and W2 represent the largest and smallest tumor diameters, respectively). Mice with tumors exceeding 15 mm in diameter were sacrificed.

### 2.6. Cell Growth Inhibition

Colon cancer cells were treated with different concentrations of 1 to 10 mM of 2-DG with and without 0.5 to 2 mM of phenformin in appropriate media for 24 to 72 h. Cell proliferation was measured with the 3-(4,5-dimethylthiazol-2-yl)-2,5-diphenyltetrazolium bromide (MTT; Sigma) assay, and the resulting formazan dye color was measured at 570 nm as previously described [23]. For the assessment drug response of colon tumor organoids, dissociated organoids with TrypLE Express were mixed in MBM  +  Matrigel (1:3 ratio) and seeded onto 96-well white plates (10 μL of 2 × 10^3^ cells per well; Corning). After gelation, 100 µL MBM was added to each well. After growing for 3–4 days, the organoids were treated with docetaxel, phenformin with 2DG, or doxorubicine for 7 days in triplicate, and the medium with fresh drug was changed every 2 days. Cell viability was measured using MTT assay as well as a luminescent CellTiter-Glo (Promega, Madison, WI, USA) kit.

### 2.7. Flow Cytometric Analysis through Annexin V and Propidium Iodide Double Staining

For Annexin V/Propidium iodide (PI) assays, cells were stained with Alexa Fluor 488 Annexin V and PI, and apoptosis was evaluated via flow cytometry according to the manufacturer’s protocol (Invitrogen). Briefly, 2 × 10^5^ cells were stained with 5 μL of Annexin V-FITC and 1 μL of PI (100 μg/mL) in 1× binding buffer (50 mM HEPES, 700 mM NaCl, 12.5 mM CaCl_2,_ pH 7.4) for 15 min in the dark. The number of apoptotic cells was determined with a FACSCanto II flow cytometer (BD Biosciences, Meylan, France). Both early apoptotic (Annexin V-positive, PI-negative) and late apoptotic (Annexin V-positive and PI-positive) cells were included in cell death determinations.

### 2.8. Statistical Analysis

Data are expressed as the means ± SD of triplicate samples or at least three independent experiments. GraphPad Prism (V9) or Excel 2016 ver.1.0 (Microsoft) software packages were utilized to determine statistical significance using Student’s *t*-test and results with a *p*-value of <0.05 were considered statistically significant.

## 3. Results

Co-treatment of phenformin and 2-deoxy-glucose induced cell growth inhibition on colon adenocarcinoma cells. It would be valuable to prove the fundamental working mechanism of the antitumor effect of chemotherapeutics on colon adenocarcinoma cells. Previously, we observed the antitumor effects of phenformin and 2-deoxy glucose (2-DG) on p53 gain-of-function mutant tumors via genetically modified mice models as well as human lung cancer cells [23]. To confirm the antitumor activity in different tumor types, we tested the chemotherapeutic effect on three genetically unique colon adenocarcinoma cell lines; HT29 cells contain both *Braf^V600E^* and *Tp53^R175H^* mutations, SW600 cells contain *Tp53^R175H^*, and RKO cells has no mutation on these two points. We first examined the efficiency of phenformin and 2-DG on colon cells as potential therapeutics. When cell growth inhibition was analyzed in the MTT assay, each cell line showed growth inhibition to each drug and was synergistically responsive to the combinational treatment of two metabolic inhibitors, regardless of genetic variations; HT29, and SW620 cells showed dramatic growth inhibition, as low as 20% and 24%, respectively, after 3 days of drug treatment (Figure 1). Combinational administrations were calculated via CompuSyn (ver.1.0, ComboSyn Incorporated., Dr. Dorothy Chou) for HT29 cells. Synergistic effects were exhibited in 10 cases of dose combination among 15 cases tested, such as, 2 + 0.5, 2 + 1, 5 + 0.5, 5 + 1, 5 + 2, 10 + 0.1, 10 + 0.5, 10 + 1, 10 + 2, 10 + 5 for 2DG (mM) + phenformin (mM), as CI values were revealed to be <1 (Appendix A). These results indicate that combination treatment with phenformin and 2-deoxy glucose efficiently inhibited colon adenocarcinoma cell growth.

Phenformin and 2-deoxy glucose reduced tumor growth of HT29 cell xenograft. As the growth inhibitory efficacy of phenformin and 2DG was uncovered on different genotyping colon tumor cells, we intended to evaluate the effects on in vivo xenografts. Generated subcutaneous tumors from HT29 cells containing both *Braf^V600E^* and *Tp53^R175H^* mutations were gradually shrunk up to an average 60% of their original size after 3 weeks of treatment, while the deteriorating effect was barely observed on body weights (Figure 2). The antitumor effect of the combinational treatment of two drugs was then proved in vivo as well as in vitro.

Co-treatment of phenformin and 2-deoxy-glucose induced cell cycle arrest and apoptosis on colon adenocarcinoma cells. We then explored underlying cell death mechanisms upon treatment of 2DG and phenformin on colon tumor cells. After two days’ treatment with the drugs, the expression of proteins related to cell death on HT29 cells was first investigated. Cleaved caspase-3 and cleaved PARP were detected after treatment of 2 mM phenformin, with or without 1 or 5 mM 2DG; cyclinD1 expression was decreased after co-treatment of 2 mM 2DG and 0.5 or 2 mM phenformin; moreover, autophagy indicator LC3-II (17KD) expression was increased upon treatment of 2 mM phenformin and co-treatment of 1 mM or 5 mM 2DG and 0.5 or 2 mM phenformin (Figure 3A). These results indicate that multiple mechanisms orchestrate to induce cell death with co-treatment of 2-DG and phenformin. As reported in previous research [23], mutant p53 was destabilized, and the expression of phosphorylated ERK was decreased (Figure 3A), indicating cell survival signaling, such as MAPK signal transduction, may be involved in regulating cell proliferation upon co-treatment of 2-DG and phenformin. To further the exploration of related mechanisms, the changes of transcripts were analyzed. As LC3-II (17KD) expression was increased, molecules related to autophagy were assessed through reverse transcription polymerase chain reaction (RT-PCR) assay. Both ATG5 and ATG7 were increased with 5 mM of 2DG and 2 mM of phenformin treatment, although the incensement of ATG5 and beclin1 were compensated with combinational treatment compared to 2DG treatment alone (Figure 3B). The molecules involved in tumor metabolism as well as epithelial-mesenchymal transition (EMT) pathways were also analyzed. Two chemotherapeutics efficiently reduced the expressions of hexokinase-2 (HK2) and 6-phosphofructo-2-kinase/fructose-2,6-bisphosphatase-3 (PFKFB3) transcripts up to 50% (Figure 3C,D), and the expressions of many transcripts on EMT pathways were dramatically decreased, such as snail1, slug, and zeb1 (Figure 3C).

To confirm cell cycle arrest and/or apoptotic cell death after co-treatment of the drugs, FACS analysis was also performed after 48 h treatment of two drugs with different combinations (Figure 4A). A total of 80% of HT29 cells were arrested at G1, and only 10% of cells were on S phase when cells were treated with 5 mM 2-DG and 2 mM phenformin (Figure 4B). Approximately 25% of the HT29 cells experienced apoptosis after being treated with 1 mM 2-DG and 2 mM phenformin or 5 mM 2-DG and 2 mM phenformin (Figure 4C). Together, co-treatment of 2-DG and phenformin decreased cell proliferation through cell cycle arrest and apoptosis, and maybe autophagy, indicating these metabolic inhibitors can be repurposed as anticancer therapeutics.

Co-treatment of phenformin and 2-deoxy-glucose reduced the growth of the patient-derived xenograft colon tumor. To evaluate antitumor effects on patient-derived models, we first performed co-treatment of two metabolism inhibitors on previously established colon cancer patient-derived xenograft (PDX) models in vivo. PDX models contain *Braf^V600E^* and *Tp53^R175H^* mutations. When treated with phenformin and 2-DG for three weeks, and once the tumor size had reached approximately 100 mm^3^, the tumor growth of the drug-treated mice had obviously slowed down, to 30 to 40% of the tumor growth in PBS-treated mice (Figure 5, PDX1; Figure 6, PDX2). Additionally, no body weight loss was observed in any of the PDX mice in each group (Appendix A). These results indicate co-treatment of phenformin and 2-DG efficiently reduced the growth of colon tumors.

The growth of established colon tumor organoids was retarded with the co-treatment of phenformin and 2-deoxy-glucose. We generated 3D cultured colon adenocarcinoma patient-derived organoids, which have recently been appreciated as one of the best patient-derived models (Table 1). Cultured colon tumor organoids showed different morphology, such as filled-round, thin-layered hollow, or thick-layered hollow shapes (Figure 7A, [24]). The antitumor efficiencies for the co-treatment of phenformin and 2DG, doxorubicine, or taxotere, for 72 h were evaluated for four kinds of colon tumor organoids: C70, C73, C80, and C81. Among three different chemotherapeutic treatments, the co-treatment of phenformin (0.1 mM) and 2-deoxy glucose (5 mM) was the most efficient for all the colon organoids (Figure 7B). C81 responded to the treatment with a less than 50% growth rate after 3 days of treatment; however, C73 responded the least to the drugs. Taxotere reduced the C80 organoid growth as efficiently as a co-treatment of phenformin and 2-DG.

To confirm the co-treatment efficiency of the two metabolic inhibitors, we treated resistant C73 organoids with various combinations of phenformin and 2DG, including different concentrations of each drug, as well as individual or co-treatments. For C73 organoids, considering they had the least reduction in growth rate, and given the characteristic 3D-culture of the organoids, we extended the treatment up to 4 days. Co-treatment with the two drugs reduced cell growth to 85% of the normal growth rate (Figure 8A) and the morphology of the organoids shrunk or had dark-filling (Figure 8B), although treatment with each drug only sustained the growth. Taken together, a decrease of cell growth with the co-treatment of metabolism inhibitors was validated on patient-derived colon tumor organoids.

## 4. Discussion

Recent studies have indicated that targeting the aberrant cancer metabolism has an enhanced antitumor effect on various tumor types [28]. Previously, we confirmed the antitumor effect of phenformin with 2DG in lung cancer cells, which induces glucose-depletion [29] and has been reported to consolidate lactate acidosis [21]. Here we report that phenformin exerted synergistic lethality on colorectal cancer cells and xenografts in a low-glucose environment due to glucose analogue, 2DG, treatment. The combinational treatment of the drugs also promoted antitumor effects on patient-derived xenografts and patient-derived organoids.

The cell death mechanism was related to cell cycle arrest, apoptosis, and autophagy in HT29 colon adenocarcinoma cells by reducing the detrimental gain-of function mutant p53 protein and phosphor-Erk. The effect of metformin or phenformin was previously demonstrated in several chemo-resistant colorectal cancer cells [10], in which apoptosis and the disturbance of EMT pathways were deeply involved. For that, signal transducer and activator of transcription 3 (stat3) and transforming growth factor β receptor 2 (TGFBR2) played critical roles. In our studies, phenformin and 2DG inhibited the growth of colorectal cancer cells via several cell survival pathways, especially decreasing the expression of p-Erk. Additionally, the combinational treatment intervened in glucose metabolism and EMT, thereby resulting in decreased expressions of major molecules such as *HK2*, *PFKFB3*, *snail*, and *zeb1*. *PFKFB3*, together with *HK2*, play major roles in glycolysis that are occasionally overexpressed in aberrant tumor microenvironments (TMEs) [30]. Ongoing and further molecular investigations utilizing patient-derived models, such as tumor organoids, would uncover relevant signaling pathways and molecules for the drug working mechanism.

Not only using xenograft in vivo analysis of HT29 cells, the antitumor efficacy of these chemotherapeutics were confirmed in patient-derived models, PDXs, and colorectal tumor organoids, providing a novel strategy for the treatment of chemo-resistant rectal cancer. With the intention of exploring drug specificity on genomic alteration, in vivo PDX experiments were performed with models containing two driver mutations for colorectal cancers, *Braf^V600E^* and *Tp53^R175H^*. The antitumor efficacy of the combinational treatments was also proved on PDX1 harboring both mutations, in agreement with the HT29 xenograft in vivo assays.

The efficacy of phenformin alone was validated in various cancer types, mainly using established cancer cell lines [4,5,6,10,31,32]; however, fewer investigations were reported for its effect in combination with other agents. Reports for the combination treatment of 2DG and phenformin demonstrated the increased antitumor efficacy compared to a single agent application [21]. The sequential treatment of phenformin and mTOR inhibitors strikingly led to a dramatic improvement in the survival of orthotopic xenograft mice of human hepatocellular carcinoma (HCC) [33]. Validation of the efficacy of phenformin in the PDX model of pancreatic cancer has also accumulated, encouraging it for development as a promising antitumor medication [7]. Likewise, the precise evaluation of phenformin’s efficacy, alone or in combination with other therapeutic strategies, would be crucial to using models with fidelity.

Three-dimensional tumor organoids have been successfully cultivated from various types of cancer patients [34,35,36] as the culture system from intestinal epithelial organoids was successfully established in 2009 [37,38]. The cultured tumor organoids have been appreciated as promising tumor models due to the fidelity of patient tissues. The value of building up the cancer biobank originating from various cancer tissues has been proven as an infinitely useful platform for tumor modeling as well as antitumor chemotherapeutics screening. In the efforts toward cancer bio-banking, we have successfully generated hundreds of tumor organoids, including lung and colon tumors [35]. In this study, the effect of chemotherapeutics disturbing metabolism was also tested on organoids from four colon adenocarcinoma patients, and the efficacy of the co-treatment of phenformin and 2DG was verified on colon organoids. Together with results from patient-derived xenografts, the effect of the drugs evaluated on established colorectal cell lines in vitro and in vivo was further confirmed with patient-derived models, strongly indicating their potential as anticancer therapeutics. Systematic genomic bioinformatics utilizing patient-derived models would provide mechanistic knowledge for a treatment strategy for colorectal cancers.

Although the FDA has withdrawn the drug since 1978, and cases of phenformin-induced lactic acidosis have continued to be reported, phenformin has still been legally available in several nations, such as Italy, China, Greece, and Portugal. Considering its powerful antitumor abilities, phenformin deserves to be further investigated in combination with chemo and/or targeted therapy to treat cancer patients. In this regard, a combination of 2DG and phenformin may be a promising candidate for synergistic antitumor treatment.

## 5. Conclusions

Disturbing tumor metabolism with chemotherapeutics such as phenformin and 2-deoxyglucose deteriorates the growth of colorectal tumors with various genetic characteristics. The results indicate that phenformin and 2-deoxyglucose, by inducing apoptosis and interrupting tumor metabolism, are potential antitumor drugs for the treatment of colorectal tumors.

## Figures and Tables

**Figure 1 cells-12-02859-f001:**
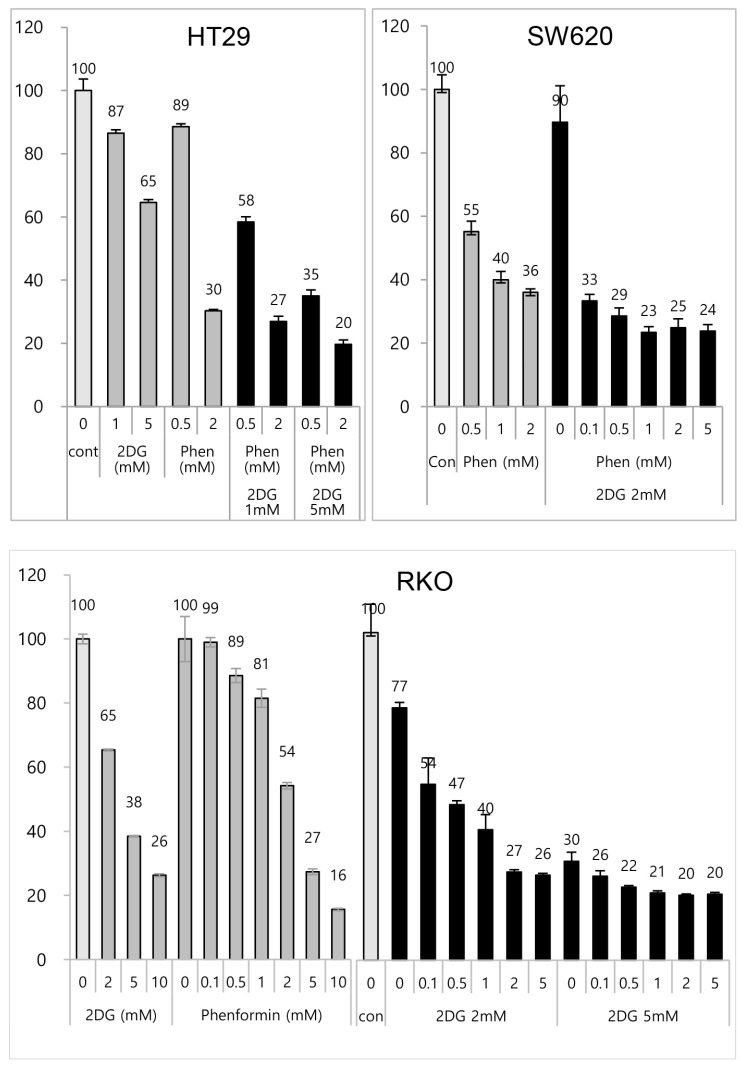
Effect of phenformin and 2-deoxyglucose was evaluated on colon cancer cell growth in vitro. Colon adenocarcinoma cells, HT29, SW620, and RKO cells, were treated with increasing concentrations of 2-DG (1 mM, 2 mM, 5 mM or 10 mM) and/or phenformin (0.1 mM, 0.5 mM, 1 mM, 2 mM, or 5 mM) for 48 h. Viability was determined via MTT assay. The absorbance was measured with an ELISA plate reader (Molecular Devices) at a wavelength of 540 nm, and the inhibitory effects were normalized to untreated condition.

**Figure 2 cells-12-02859-f002:**
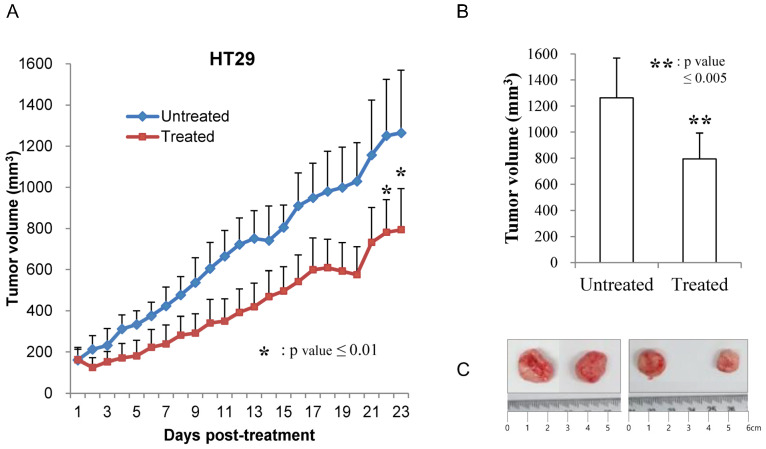
The effect of phenformin and 2-deoxyglucose was evaluated on in vivo xenografts using HT29 colon cancer cells (**A**). Cells 2 × 10^6^ formed tumors approximately 100 mm^3^ in size 12–14 days after subcutaneous inoculation. Mice were then treated orally with PBS (vehicle, denoted as untreated) or 750 mg/kg phenformin and 2DG (denoted as treated) once a day for 21 days. The tumor volume was calculated using the formula for an ellipsoid sphere: W1 × W2 × W2/2 = x mm^3^, where W1 represents the largest tumor diameter and W2 the smallest tumor diameter. Independent xenograft experiments were performed three times. * *p* < 0.01 refers to comparisons between drug- and vehicle-treated tumors (**B**). The average size of harvested tumors was graphed at the end of the experiments and the harvested tumors were represented (**C**).

**Figure 3 cells-12-02859-f003:**
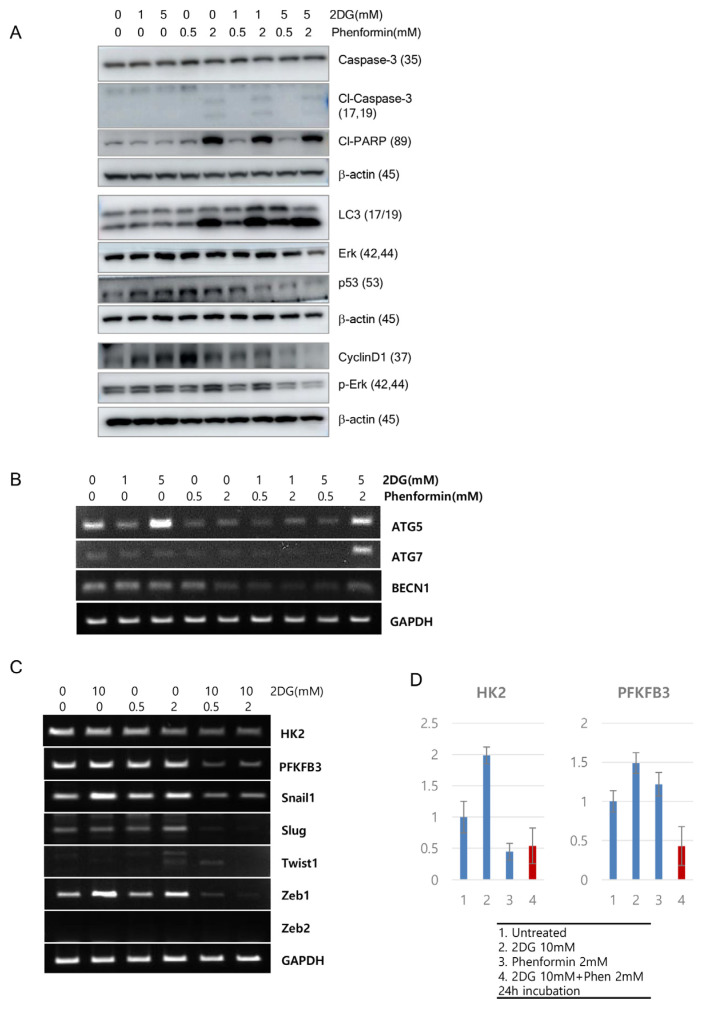
Cell death mechanism was evaluated for the effect of 2DG and phenformin for the inhibition of HT29 colon adenocarcinoma cells. (**A**) The protein expressions were analyzed after treatment with 2DG and phenformin on western blotting. Cells were incubated in the presence of different concentrations of 2DG and/or phenformin for 48 h. Whole-cell lysates (25 μg) were separated on gels and blotted to antibodies against Caspase3, PARP, cyclinD1, LC3, p53, phosphor-ERk, total ERK, and β-actin. (**B**) Transcripts were assessed by reverse-transcription PCR assay for the molecules involved in epithelial-mesenchymal transition (EMT) pathways. (**C**) The transcripts involved in cancer metabolism and cancer stem cells were assessed by reverse-transcription PCR assay. (**D**) The deceased expressions of HK2 and PFKFB3 were graphed in different conditions.

**Figure 4 cells-12-02859-f004:**
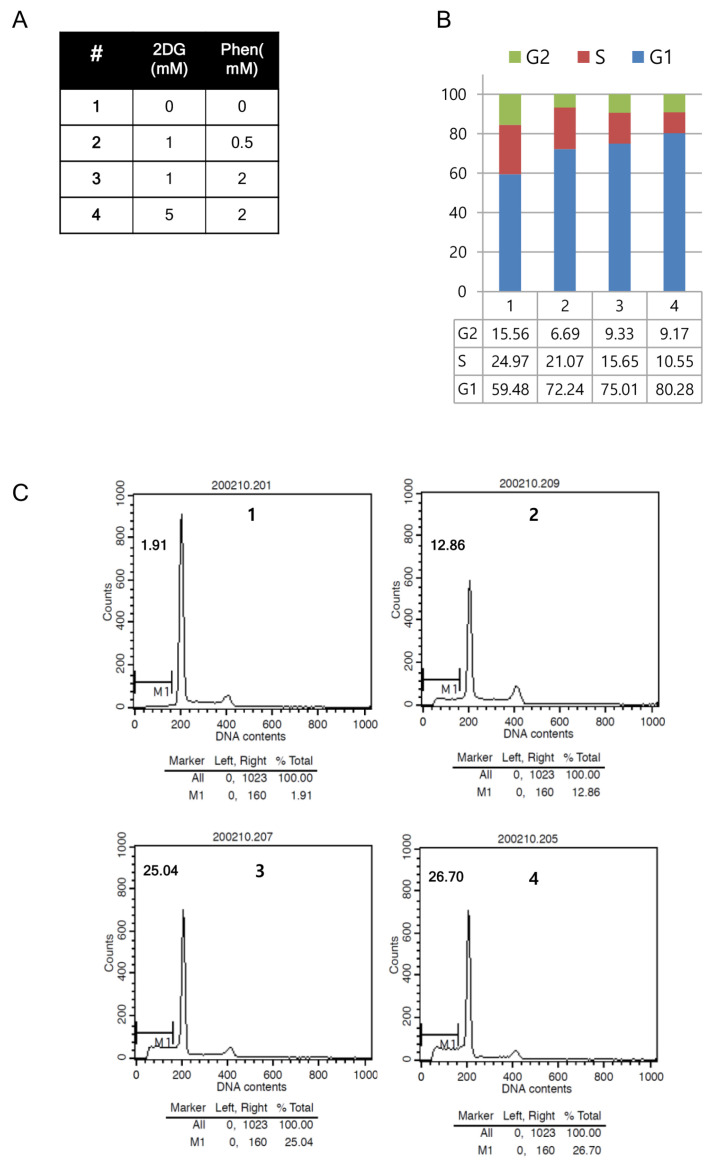
(**A**) FACS analysis was carried out on HT29 cells after co-treatment of different combination of 2DG and/or phenformin for 48 h. (**B**) The graph depicted as a percentage of cell populations in G1, S, and G2 phase of the cell cycle. (**C**) Cells in subG1 (apoptotic cells) were denoted as M1 on the graph after treatment of 2DG and phenformin.

**Figure 5 cells-12-02859-f005:**
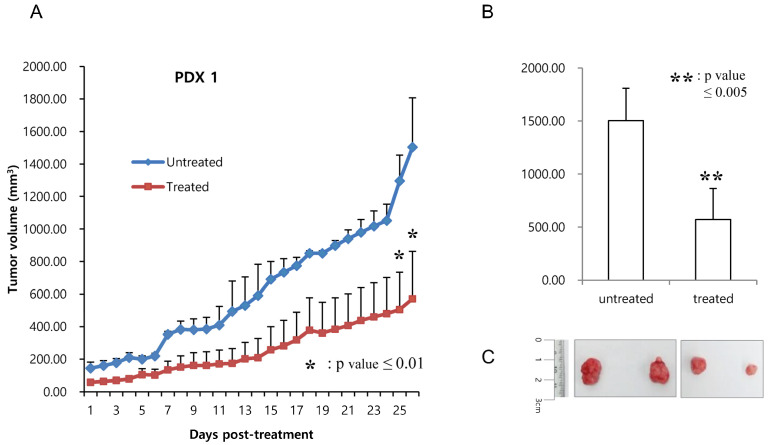
The effect of 2-DG and phenformin on tumor growth was evaluated in two patient-derived xenografts (PDX1). (**A**) Tumors reached approximately 100 mm^3^ in size at 30–40 days after inoculation. Mice were then treated orally with PBS (untreated) or 750 mg/kg phenformin and 2-DG (treated) once a day for 21 to 24 days. Tumor volume was calculated using the formula for an ellipsoid sphere: W1 × W2 × W2/2 = x mm^3^, where W1 represents the largest and W2 the smallest tumor diameter. Independent xenograft experiments were performed three times. * *p* < 0.01 and ** *p* < 0.005 in comparison with vehicle-treated tumors. Tumor growth was graphically depicted according to size. (**B**) Tumors from PBS- (untreated) or drug-treated (treated) mice were collected at the end of the experiment. (**C**) The harvested tumors.

**Figure 6 cells-12-02859-f006:**
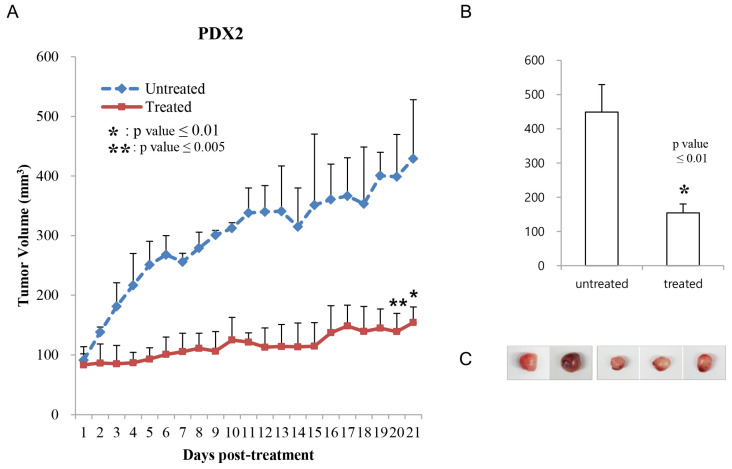
Effect of 2-DG and phenformin on tumor growth was evaluated in patient-derived xenografts (PDX2). (**A**) Tumors reached approximately 100 mm^3^ in size at 30–40 days after inoculation. Mice were then treated orally with PBS (untreated) or 750 mg/kg phenformin and 2-DG (treated) once a day for 21 to 24 days. Tumor volume was calculated using the formula for an ellipsoid sphere: W1 × W2 × W2/2 = x mm^3^, where W1 represents the largest and W2 the smallest tumor diameter. Independent xenograft experiments were performed three times. * *p* < 0.01 and ** *p* < 0.005 in comparison with vehicle-treated tumors. Tumor growth was graphically depicted according to size. (**B**) Tumors from PBS- (untreated) or drug-treated (treated) mice were collected at the end of the experiment. (**C**) The harvested tumors were represented and animal body weights were monitored during the experiment (Appendix A).

**Figure 7 cells-12-02859-f007:**
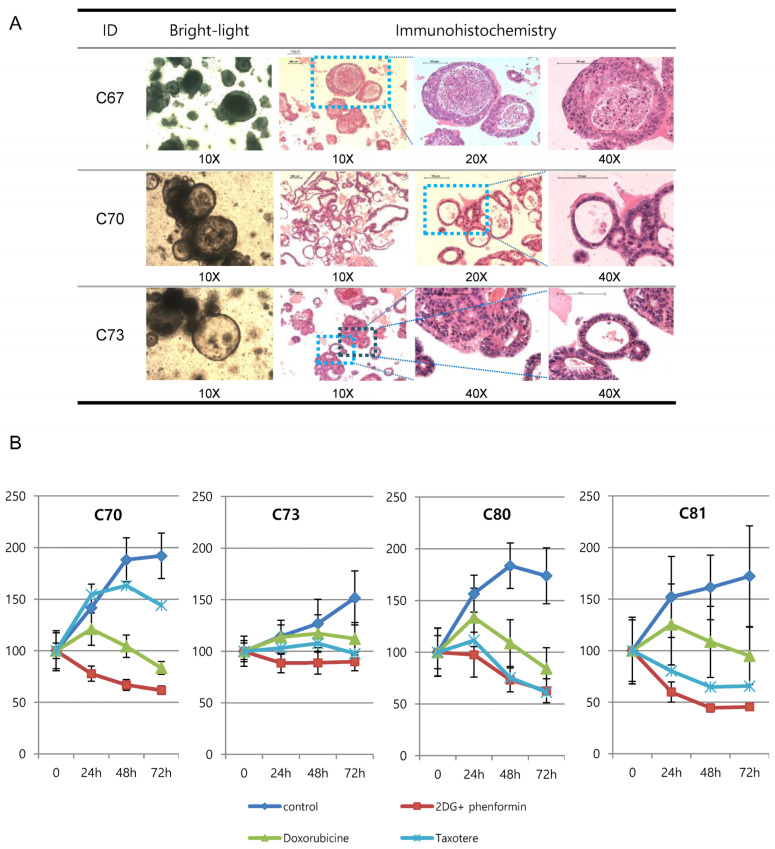
(**A**) Bright-field microscopy images and H&E staining images of colorectal patient-derived organoids. The enlarged images of colon organoids as a platform for predicting therapeutic responses. (**B**) Dose-response curves after 3 days of treatment with various chemotherapeutics. The cell viability was measured via luminescence signal intensities. The indicated drug concentrations were used to treat four colon organoids. Representative viability curves were generated from luminescence signal intensities.

**Figure 8 cells-12-02859-f008:**
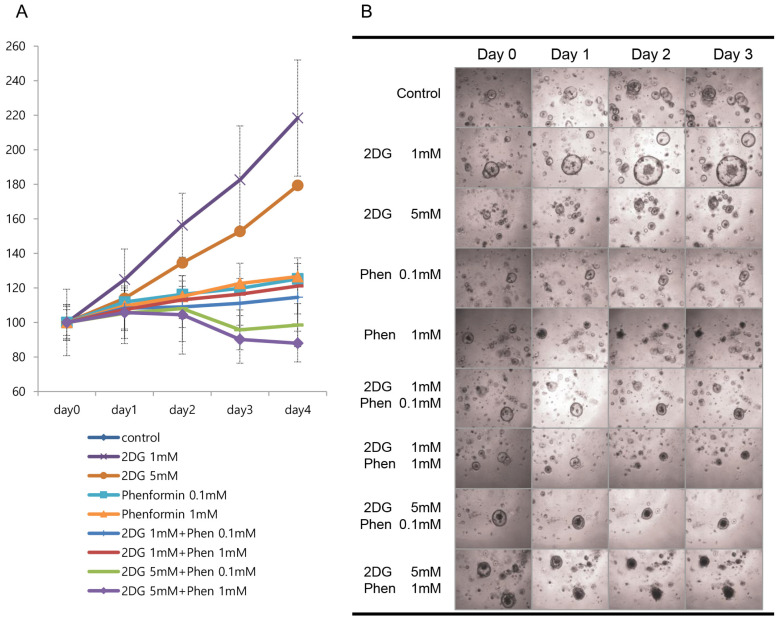
Colorectal tumor organoids were treated with various concentration combinations of two metabolic inhibitors, such as phenformin and 2DG. (**A**) Representative viability curves were generated after 4 days of treatment with chemotherapeutics. The cell viability was measured via luminescence signal intensities. (**B**) Representative bright-field microscopic images of the colorectal organoids with or without drugs showing the difference (magnification, 10×).

**Table 1 cells-12-02859-t001:** Patient information of the organoids that were generated.

				IHC		Mutation Analysis	Chemo-Therapy	Follow-Up
	Sex Age	Stage	Met ^(1)^	p53	EGFR	Kras	FOLFOX +Ava	2 Years
C67	M 58	pT4aN2b	LN ^(2)^	-	+++	WT ^(4)^	Treated	NA
C70	F 39	pT1N0Mx		+++	NA ^(3)^	WT	NT ^(5)^	Normal
C73	M 60	pT2N0		+++	NA	WT	NT	Adenoma
C80	M 47	pT3N2b	Liver	-	++	WT	Treated	lung met
C81	F 53	pT3N2bM1b	LN	-	+++	G12D	Treated	NA

Met ^(1)^, metastasis was detected; LN ^(2)^, lymph node; NA ^(3)^, not available; WT ^(4)^, wild type (not mutated); NT ^(5)^, not treated. -, no expression, ++, medium expression, +++, strong expression.

## Data Availability

Data are contained within the article.

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
