# Peer review of "Repurposing Metabolic Inhibitors in the Treatment of Colon Adenocarcinoma Patient-Derived Models"

_cells, 2023, doi:10.3390/cells12242859_

Round 1
Reviewer 1 Report
Comments and Suggestions for Authors
This manuscript investigated the effect of metabolic inhibitors along with biguanides in the treatment of colon adenocarcinoma patient-derived models. Overall, the manuscript is very well written and provides interesting results. While the findings are valuable, implementing the following comments would add to the interpretation of the study.
Major comments
Introduction
1. The references bellow should be cited.
https://doi.org/10.1016/j.bbrc.2022.12.069
https://doi.org/10.1016/j.ejca.2022.02.005
https://doi.org/10.3390/biomedicines10050955
2. The manuscript needs minor English revision.
3. The sentence “Our findings suggested that metabolic inhibitors represent effective therapeutic agents for the treatment of colon tumor” should be omitted from introduction unless suggested by the journal.
4. How were the doses selected?
5. In the results section, it is written that phenformin +2DG had synergistic effects. How was this effect calculated?
6. Flowcytometry data of apoptosis should be provided.
7. The discussion section is rather short. It is suggested that the results be discussed more in detail.
Comments on the Quality of English Language
The manuscript needs minor English editing.
Author Response
This manuscript investigated the effect of metabolic inhibitors along with biguanides in the treatment of colon adenocarcinoma patient-derived models. Overall, the manuscript is very well written and provides interesting results. While the findings are valuable, implementing the following comments would add to the interpretation of the study.
Major comments
Introduction
- The references bellow should be cited.
https://doi.org/10.1016/j.bbrc.2022.12.069
https://doi.org/10.1016/j.ejca.2022.02.005
https://doi.org/10.3390/biomedicines10050955
The references the Reviewer recommended were cited in proper places in the text; Introduction section in page2.
- The manuscript needs minor English revision.
Although we were, fortunately, able to revise ‘Introduction’ and Discussion’ sections in the manuscript after discussing with our colleague professor who is an American, we never mind to have an editorial revision from the journal if necessary.
- The sentence “Our findings suggested that metabolic inhibitors represent effective therapeutic agents for the treatment of colon tumor” should be omitted from introduction unless suggested by the journal.
As the Reviewer recommended, the sentence was omitted from introduction.
- How were the doses selected?
Treatment doses were selected based on our previous studies for lung and pancreatic cancer cells (ref #23, figure2B and unpublished data). According to the previous studies, IC50s of mouse lung cancer cells varied from 2.4mM to 66mM for 2DG, 0.1mM to 2mM for phenformin, and the ratio of 2DG and phenformin was 10:1 to 20:1 for synergistic effect, respectively. The adequate doses and combination ratio should be diverse depending upon the cell types, and we treated colorectal cells based on previous studies.
- In the results section, it is written that phenformin +2DG had synergistic effects. How was this effect calculated?
We explained about the doses in detail in the Results section; page4. Adequate concentrations for combinational treatment were mentioned to be >1 of CI index according to CompuSyn analysis (supplementary figure1). When cells were treated with various combinations, such as , 2+0.5, 2+1, 5+0.5, 5+1, 5+2, 10+0.1, 10+0.5, 10+1, 10+2, 10+5 for 2DG (mM)+phenformin (mM), synergistic effects were observed.
- Flowcytometry data of apoptosis should be provided.
FACS analysis was performed and shown in figure4C. Although subG1 is acceptable to represent apoptotic cells in many cases, we are able to repeat FACS experiments if necessary.
- The discussion section is rather short. It is suggested that the results be discussed more in detail.
We are totally encouraged with this critical recommendation and discussed more and in detail for not only our results but also peer-reviewed papers. Please consider our revisions for ‘Discussion’ section, hoping this report suggests some envision to treat colorectal tumors.
Reviewer 2 Report
Comments and Suggestions for Authors
The present paper deals with investigation of synergistic antitumor efficiency of two metabolic inhibitors – phenformin and 2-deoxyglucose, against colon cancer models. First, phenformin and 2-DG were tested against 3 cancer cell lines, separately and synergistically, at various concentrations. Subsequently, the in vivo experiment was performed with HT29 xenograft in mice, with oral admission of drugs. Synergistic antitumor effect of two drugs via apoptosis of cancer cells have been confirmed by reduction of mean volume of harvested tumors and western blot analysis. The effect of novel drug combination has been compared to common antitumor drugs on the patient-derived organoids, using the light microscopy and immunohistochemistry methods.
The manuscript is interesting, well conceptualized and brings significant results. There are a few issues to further improve the quality of the manucsript:
1. The debate on phenformin repurposing should be placed in Introduction, rather than Discussion section. The reasons of its retrieval as antidiabetic and the main arguments for its repurposing as anticancer drug, with corresponding references, should be mentioned. Similarly, there is almost nothing on 2-DG use as antitumor drug in Introduction. The reasons for selecting the investigated combination of drugs should be mentioned and shortly discussed.
2. On the other hand, Discussion section lacks explicit representation of crucial data that can be derived from Results. Which combination has brought the best antitumor efficiency? How much the efficiency was improved compared to isolated phenformin and/or 2-DG, and/or other reference drugs? Which are the key points of action of two drugs in synergy – does the mechanism of cell apoptosis differ compared to isolated drugs?
3. Figures and legends are seemingly too large and contain too much panels per figure, especially in Figure 3. Please, where possible, consider representing graphs as separate images with separate subtitles, as they would be much easier to read. Also, consider transfering some panels to text, supplementary data or tables in order to reduce number of pictures and improve clarity (e.g. Figure 1 B2, B4; Figure 2 C, Figure 3 A2, A4, B2, B4 )
4. One minor question – the tumor volume was calculated from an approximate formula. If excision of tumors was performed and they were solid, have you tried to measure the volume directly in a vessel with liquid, by measuring the raise of liquid level.
Author Response
Comments and Suggestions for Authors
The present paper deals with investigation of synergistic antitumor efficiency of two metabolic inhibitors – phenformin and 2-deoxyglucose, against colon cancer models. First, phenformin and 2-DG were tested against 3 cancer cell lines, separately and synergistically, at various concentrations. Subsequently, the in vivo experiment was performed with HT29 xenograft in mice, with oral admission of drugs. Synergistic antitumor effect of two drugs via apoptosis of cancer cells have been confirmed by reduction of mean volume of harvested tumors and western blot analysis. The effect of novel drug combination has been compared to common antitumor drugs on the patient-derived organoids, using the light microscopy and immunohistochemistry methods.
The manuscript is interesting, well conceptualized and brings significant results. There are a few issues to further improve the quality of the manuscript:
- (A) The debate on phenformin repurposing should be placed in Introduction, rather than Discussion section. The reasons of its retrieval as antidiabetic and the main arguments for its repurposing as anticancer drug, with corresponding references, should be mentioned. (B) Similarly, there is almost nothing on 2-DG use as antitumor drug in Introduction. The reasons for selecting the investigated combination of drugs should be mentioned and shortly discussed.
(A) We moved the debated on phenformin repurposing in Introduction section as Reviewer’s recommendation (page 1).
(B) Also, the use of 2-DG as an antitumor drug was mentioned in Introduction, especially for the beneficial effect for the combination with phenformin (page 1).
- On the other hand, Discussion section lacks explicit representation of crucial data that can be derived from Results. Which combination has brought the best antitumor efficiency? How much the efficiency was improved compared to isolated phenformin and/or 2-DG, and/or other reference drugs? Which are the key points of action of two drugs in synergy – does the mechanism of cell apoptosis differ compared to isolated drugs?
We included deeper discussions based on our experimental results in Discussion section. Also, dose-effect curve and CI values from the assessment via CompuSyn software was included in Results section and supplementary figure1 (page4) to prove synergistic effect with two chemotherapeutics
- Figures and legends are seemingly too large and contain too much panels per figure, especially in Figure 3. Please, where possible, consider representing graphs as separate images with separate subtitles, as they would be much easier to read. Also, consider transferring some panels to text, supplementary data or tables in order to reduce number of pictures and improve clarity (e.g. Figure 1 B2, B4; Figure 2 C, Figure 3 A2, A4, B2, B4)
We changed figures’ numbering as such figure1A, figure1B…, figure2A, figure2B…, and so on, to improve clarity. Each figure size was also shrunk as some were transferred to supplementary figure2 (for example, graphs for mouse body weights during each xenograft experiment), as Reviewer’s sincere consideration.
- One minor question – the tumor volume was calculated from an approximate formula. If excision of tumors was performed and they were solid, have you tried to measure the volume directly in a vessel with liquid, by measuring the raise of liquid level.
The data shown here was performed as generally acceptable methodology for exploration of drug sensitivity in vivo, although we now realized that the recommended protocol would be precise and convenient for measuring the volume of tumors at the end point of experiment.
Round 2
Reviewer 2 Report
Comments and Suggestions for Authors
The authors have properly responded to all my suggestions. Hence, I consider that the manuscript is now acceptable for publication.